# The First Plastid Genome of the Holoparasitic Genus *Prosopanche* (Hydnoraceae)

**DOI:** 10.3390/plants9030306

**Published:** 2020-03-01

**Authors:** Matthias Jost, Julia Naumann, Nicolás Rocamundi, Andrea A. Cocucci, Stefan Wanke

**Affiliations:** 1Technische Universität Dresden, Institut für Botanik, 01062 Dresden, Germany; 2Instituto Multidisciplinario de Biología Vegetal, Universidad Nacional de Córdoba, CONICET, FCEFyN, Av. Vélez Sársfield 1611, Córdoba X5016GCA, Argentina

**Keywords:** Piperales, Hydnoraceae, *Hydnora*, *Prosopanche*, parasitic plants, holoparasite, plastid genome

## Abstract

Plastomes of parasitic and mycoheterotrophic plants show different degrees of reduction depending on the plants’ level of heterotrophy and host dependence in comparison to photoautotrophic sister species, and the amount of time since heterotrophic dependence was established. In all but the most recent heterotrophic lineages, this reduction involves substantial decrease in genome size and gene content and sometimes alterations of genome structure. Here, we present the first plastid genome of the holoparasitic genus *Prosopanche,* which shows clear signs of functionality. The plastome of *Prosopanche americana* has a length of 28,191 bp and contains only 24 unique genes, i.e., 14 ribosomal protein genes, four ribosomal RNA genes, five genes coding for tRNAs and three genes with other or unknown function (*acc*D, *ycf*1, *ycf*2). The inverted repeat has been lost. Despite the split of *Prosopanche* and *Hydnora* about 54 MYA ago, the level of genome reduction is strikingly congruent between the two holoparasites although highly dissimilar nucleotide sequences are observed. Our results lead to two possible evolutionary scenarios that will be tested in the future with a larger sampling: 1) a Hydnoraceae plastome, similar to those of *Hydnora* and *Prosopanche* today, existed already in the most recent common ancestor and has not changed much with respect to gene content and structure, or 2) the genome similarities we observe today are the result of two independent evolutionary trajectories leading to almost the same endpoint. The first hypothesis would be most parsimonious whereas the second would point to taxon dependent essential gene sets for plants released from photosynthetic constraints.

## 1. Introduction

In photoautotrophic plants, the plastid chromosome encodes essential genes for major photosynthesis related functions and is mostly considered highly conserved with respect to gene order and gene content [1] as well as its organization as quadripartite structure consisting of two single copy regions separated by two copies of an inverted repeat [2,3]. The plastomes of parasitic plants are a prime subject to study the possibilities and changes of otherwise highly conserved genomic structures that can occur when organellar genomes are released from selective constraints [4,5]. Plastid genomes of members from nearly all of the at least 11 independently evolved parasitic angiosperm lineages [6] have been studied to a more or less extensive degree [3,7,8,9,10,11,12,13,14,15]. Recent in-depth papers and reviews summarized similar evolutionary stages of plastome decay [5,16,17], eventually resulting in possible complete loss in Rafflesiaceae [14], depending on the respective level of heterotrophy. However, a vast variety of changes in plastid genome size and organization, gene content, nucleotide composition and mutational rates has been reported within individual lineages and the speed and extent of degradation seem lineage-specific [3]. Despite those lineage-specific differences, an underlying pattern can be found that plants follow as they transition from autotrophic to heterotrophic lifestyle. With increasing heterotrophy, genes involved in photosynthesis show signatures of relaxed functional constraints and eventually become pseudogenized and lost, often starting with *ndh* genes [16,18]. Following the initial losses is a relaxation of purifying selection in genes related to photosynthesis and genes involved in the translation and transcription machinery. General housekeeping genes that carry out non-photosynthesis related functions can show less relaxation or even slightly higher purification [18]. There are many examples of the different stages of parasitism in numerous lineages with the Orobanchaceae being the prime subject to study the whole range from facultative to holoparasitism [19,20]. It is apparent that often gene order and plastome structure such as the quadripartite nature (large single copy region LSC, small single copy region SSC and inverted repeats IR) are retained in different parasitic plastomes. Yet, there are some exceptions such as the loss of the inverted repeat for example in the highly reduced plastomes of *Pilostyles* [12,21], *Balanophora* [22] and many other lineages of autotrophic plants [23,24,25,26]. Functionally the inverted repeats are hypothesized to stabilize the plastid genome [27]; therefore, their loss is shown to lead to more frequent rearrangements [28].

One of the 11 parasitic angiosperm lineages is the holoparasitic family Hydnoraceae (Piperales). In contrast to all other parasitic angiosperm lineages Hydnoraceae are the only lineage outside the monocot and eudicot radiation and among one of the oldest parasitic lineages with an estimated stem group age of ~91 MYA [29]. The family consists of the two genera *Hydnora* (7 species) [30] and *Prosopanche* (5/6 species) [31] with *Hydnora* occurring exclusively in the Old World and *Prosopanche* in the New World [32,33]. According to molecular dating analyses the two genera split from each other about 54 MYA [29]. The plastid genomes of Hydnoraceae are also among the least known with a single known plastome of *Hydnora visseri* [10], for which not only a drastic reduction in genome size but also in gene content has been shown, exceeding the loss of genes involved in the photosynthesis apparatus, yet maintaining the quadripartite structure of the plastome as well as showing nearly identical gene order and orientation compared to the closely related photoautotrophic genus *Piper* (Piperaceae, Piperales) [10]. Given that the two Hydnoraceae genera diverged in the early Eocene and the plastome of *Hydnora visseri* is among the most reduced holoparasitic angiosperms knowledge about its sister genus *Prosopanche* would be highly valuable for putting the sequenced plastomes in context to better understand the evolution of this specific lineage and to compare it to the other different parasitic lineages. We here report the plastid genome of *Prosopanche americana* and compare it to the published genome of *Hydnora visseri* as well as to the plastome of *Aristolochia contorta* [34] (Aristolochiaceae, the closest autotrophic relatives of Hydnoraceae).

## 2. Results

### 2.1. The Plastome of Prosopanche Americana

In total, 373 million reads were sequenced and de novo assembled using CLC Workbench [35], resulting in a total of 8,342,403 scaffolds of which 372 had BLAST hits for plastid features (Figure 1), using a query of 45 angiosperm plastomes from GenBank and setting the e-value to 1e-10. In depth analyses of these scaffolds based on quality of BLASTn hit in combination with scaffold coverage, resulted in the exclusion of all but scaffold 424, which received the highest number of BLAST hits for plastid genome features. Additionally, it is the longest scaffold with plastid BLAST hits (28,191 bp) and shows the highest depth of coverage (4678) and with a total of 952,705 reads mapped to it. The scaffold has identical ends in sequence, forming 47 bp of overlap, allowing for circularization. The circularization was verified by PCR and the Sanger sequenced product was aligned to the assembled scaffold, which also revealed and corrected a misassembly and an insertion at the respective contig ends, but introduces two unresolved characters in this region.

With this, the *Prosopanche americana* plastome is 28,191 bp in length (Figure 2). Using DOGMA (% identity cutoff = 25, e-value = 1e-5) [36], 25 potential plastid genes were initially identified, consisting of 17 protein coding genes, 3 rRNAs and 7 tRNAs. Only 24 of the initial 25 genes could be verified and were used for further testing and analysis because the hit for *trn*P-GGG could not be confirmed and is a result of the low stringency search settings. With the methods described below, one additional rRNA could be identified, which was not detected by the DOGMA analysis. For comparison, we also ran an analysis of scaffold 424 using MFannot [37] which resulted in a similar result, but failed to identify *rrn*5, *ycf*2 or *trn*fM-CAU. *trn*W identified by DOGMA is identified as *trn*C with MFannot. In total, the plastome contains 25 genes, of which 24 are unique. There are 14 ribosomal protein genes (*rpl*2, *rpl*14, *rpl*16, *rpl*36, *rps*2, *rps*3, *rps*4, *rps*7, *rps*8, *rps*11, *rps*12, *rps*14, *rps*18, *rps*19), two of which contain introns (*rpl*16, *rps*12), four ribosomal RNA genes (*rrn*4.5, *rrn*5, *rrn*16, *rrn*23), five genes coding for tRNAs (*trn*E-UUC, 2x *trn*I-CAU, *trn*fM-CAU, *trn*W-CCA, *trn*Y-GUA) and three genes with other or unknown functions (*acc*D, *ycf*1, *ycf*2). The *Prosopanche americana* plastome (GenBank accession number MT075717) lacks genes involved in photosynthesis, such as genes for the photosystems I and II, RuBisCO large subunit, cytochrome-related or ATP synthase, NADH dehydrogenase genes, as well as RNA polymerase subunits, *clp*P and *mat*K. All protein coding genes in the plastome have feature continuous reading frames with the only exception of the *rps*19 gene, which does not possess an in-frame start codon in close proximity. Amino acid isotypes predicted by DOGMA [36] are confirmed by tRNAscan-SE [38,39] anticodon prediction (Appendix A) in all but the case of *trn*W-CCA. The underlying sequence is predicted to form a cloverleaf structure with three leaves but it has an ACA anticodon (as predicted by MFannot) instead of CCA. Alignments of the respective tRNAs from *Aristolochia* (*Aristolochia contorta* NC_036152) [34] indicate that this tRNA is homologous to the *Aristolochia trn*W-CCA. A three-loop cloverleaf 2D structure is predicted for all identified tRNAs, except for *trn*Y-GUA. The latter shows an additional, variable arm in the tRNAscan-SE prediction, which is also present in the *trn*Y of *Aristolochia contorta*. The anticodon versus isotype model prediction (Appendix A) by tRNAscan-SE is consistent for two tRNAs (*trn*fM-CAU and *trn*Y-GUA). The correctness of the anticodon prediction for all tRNAs was verified via alignments with respective genes of *Aristolochia contorta*. Given the abovementioned information, it is likely that *trn*W-CCA is a pseudogene.

The *Prosopanche* plastome contains 27 intergenic regions with an average length of 173 bp and the longest one being 879 bp in length. In total, 23 open reading frames were identified within the intergenic regions (ORF finder, implemented in Geneious 11.1.5 [40]), while setting the minimum size to 90 bp, which is the shortest plastid encoded protein coding gene in Piperales and allowing for the detection of smaller ORFs within a larger ORF. In-depth analyses of intergenic regions through BLAST or automatic annotation (as low as 50% sequence similarity) did not return meaningful hits for genes or pseudogenes. Structurally, the plastome of *Prosopanche americana* does not have any inverted repeats and therefore does not show the typical quadripartite structure. All genes are found as a single copy on the plastome, except the *trn*I-CAU gene, which is found in two copies both showing the same orientation on the circular plastome. Analysis of the flanking regions of both copies (Figure 3) show a stretch of 195 bp with only four substitutions between them. In addition to the fully duplicated tRNA, an alignment of the two copies also shows a duplication of the first 25 bp of the *rpl*2 gene. The gene is truncated downstream in one repeat (copy B) due to sequence divergence and the end of the duplication. The coverages of the duplication copies are neither significantly higher nor significantly lower than the average plastome coverage (4678). In copy A coverage ranges from 4285 to 4682 and from 4316 to 4868 for copy B.

The GC content of the whole plastome is 20.4% (Figure 2, Figure 4a). Protein coding regions, which make up 64.9% (18,303 bp) of the total length consist to 17.9% of GC nucleotides, rRNA genes make up 16.5% (4639 bp) and contain with 37.2% the highest percentage of GC in the *Prosopanche* plastome (Figure 2 and Figure 4a). Noncoding regions (excluding introns) on the other hand show the least amount of GC content with 11.3%, while making up 16.3% (4600 bp) of the plastome. 

### 2.2. Comparison of the Plastomes of the Holoparasitic Hydnoraceae

The plastome of *Prosopanche americana* (28,191 bp) is slightly larger than the plastome of *Hydnora visseri* (27,233 bp, NC_029358) even though it lacks the inverted repeat, which contributes 1466 bp to the length of *Hydnora*. *Prosopanche*, however, contains a 195 bp repeat and two additional genes (Figure 5). The remaining set of genes is shared between the genomes of *Hydnora* and *Prosopanche*. All protein coding genes are putatively functional with the exception of the *Prosopanche rps*19 and the *Hydnora rps*7. Single gene alignment of the *acc*D gene shows a drastic size difference between the copies of the two Hydnoraceae genera. The closest ATG start codon in frame eligible as *acc*D gene start of *Prosopanche* leads to a 184 bp increase in length, which does not align to the *Hydnora acc*D gene. Apart from the protein coding genes, the set of ribosomal RNAs also is identical between the sister taxa, solely the number of transfer RNAs differs between the two plastomes. *Prosopanche* contains with *trn*W and *trn*Y two additional tRNAs, yet the former in the form of a pseudogene. The gene order between the two plastomes is mostly consistent with an exception of an inversion of the ‘*rps*7-*rps*12 exon 2 and 3′- region and the ‘*trn*I-*rpl*2-*rps*2′- region. Contrary to *Hydnora visseri*, the plastome of *Prosopanche americana* does not possess inverted repeats, though it contains a duplicated *trn*I-CAU-region while both copies share the same orientation. The same tRNA, *trn*I, is the only complete gene duplicated in the inverted repeat of *Hydnora*. The overall GC content in *Prosopanche* is lower than in its sister genus (Figure 4a) with the most drastic difference showing in the non-coding regions. The dotplot (Figure A1 in Appendix B) of the two plastomes shows a fairly straight diagonal line with many interruptions due to differences on sequence level and an inversion towards the end (Figure A1, red circle) illustrating the larger of the two inversions between the species.

### 2.3. Plastome Comparison to Aristolochia

A comparison of the plastomes of Hydnoraceae with the closely related photoautotrophic *Aristolochia contorta* (NC_036152) [34] demonstrates large differences in genome size, gene and GC content. With a length of 160,576 bp, the plastome of *A. contorta* is about six times the size of the holoparasites and with a total of 131 genes it contains about six times the amount of that of *Prosopanche americana* and *Hydnora visseri*. *Aristolochia contorta* shows a characteristic plastid genome for photoautotrophic Piperales, with all necessary genes for the photosynthesis machinery [34]. The overall GC content (38.3%), as well as the GC content in different parts of the plastome, is higher than in the respective parts of the Hydnoraceae (Figure 4a) with the highest overall GC contents found in the rRNA genes across the three genera. The reduction of G and C nucleotides in *Hydnora* and *Prosopanche* occurs at a similar rate, with the latter having slightly lower percentages than *Hydnora*. The intergenic regions of *P. americana* however, show a much more drastic increase in AT content than observed in *Hydnora* and make up to 88.3% of the nucleotides (Figure 4). The comparison of changes within the three categories protein coding genes, rRNAs and intergenic regions (IGR) highlights the proportional increase in length in relation to the total plastome length for protein coding genes and the four rRNA genes (Figure 4b). The rRNA genes of the Hydnoraceae make up about three times as much of the total plastome length compared to the corresponding genes in *A. contorta*. A proportional decrease is shown for the intergenic regions in the parasitic plastomes compared to the photoautotrophic relative where IGRs contribute to about twice as much to the total plastome length.

An in-depth look at GC content and codon usage of the protein coding genes shared between the three plastomes using CodonW [41] shows a decrease in G and C nucleotides at the third codon position of these genes in the Hydnoraceae (Appendix A). The two protein coding genes with the lowest GC proportion in the parasitic plastomes are *ycf*2 and *rps*18, with *ycf*2 being also one of the longest plastid-encoded genes. The lowest GC content in *A. contorta* is found in the *ycf*1 gene with 31.9%, which is the highest percentage found in all of the *H. visseri* protein coding genes. The lowest overall G and C percentage is found in the *ycf*2 gene of *P. americana* with only 13.3%.

### 2.4. Phylogenetic Placement of Hydnoraceae

The phylogenetic maximum likelihood (ML) tree (Figure 6) of the concatenated 82 gene alignment estimated with GTR+I+G substitution model and rapid bootstrapping (1000 replicates) recovers all major clades as monophyletic with high backbone support, except for the eudicot-monocot split which has a lower bootstrap value of 68. *Prosopanche* and *Hydnora* are supported as sister genera (bootstrap value = 100) and the monophyletic Hydnoraceae are placed within the Piperales as sister to a sister group of Aristolochioideae and Asaroideae with low support (bootstrap value = 13). Piperales as a monophyletic group receive only moderate support (bootstrap value = 76). The phylogram (Figure 6, left) highlights extremely long branches, not only leading to the most recent common ancestor (MRCA) of *Hydnora* and *Prosopanche* but also from the MRCA to the terminals, demonstrating the extremely high substitution rates in Hydnoraceae.

Phylogenetic tree reconstruction based on a gene set reduced to the genes that are present in Hydnoraceae, as well as on amino acid alignments of the complete and reduced gene set recovered relationships with generally lower support that are different from widely accepted topologies (Appendix A).

## 3. Discussion

Phylogenetic tree reconstruction using maximum likelihood confidently verifies *Hydnora* and *Prosopanche* as sister genera (100 bootstrap support) and shows Hydnoraceae monophyletic, placed within Piperales. The placement of Hydnoraceae as sister to Aristolochiaceae in its widest circumscription (i.e., including Aristolochioideae and Asaroideae, Lactoridaceae not sampled) receives, however, no support (bootstrap value = 13) and might be a result of the long branches as well as missing data (few genes present in Hydnoraceae compared to other angiosperms). The inclusion of Hydnoraceae within Piperales is congruent with previous analyses. However, the positioning within the order differs between multiple different datasets and analyses. Naumann et al. [29] place Hydnoraceae as sister to *Aristolochia* and *Thottea* (Aristolochioideae) with *Lactoris* (Lactoridaceae) being sister to this group (Asaroideae not sampled). Nickrent et al. (six gene analysis) [33] reconstructed Hydnoraceae as sister to *Lactoris* (Lactoridaceae) and this group as sister to *Aristolochia*, but missing Asaroideae in the sampling. Massoni et al. [43] (12 gene analysis) recover Hydnoraceae as sister to *Lactoris* and Aristolochioideae with moderate support and this group sister to Asaroideae. 

The plastome of the holoparasite *Prosopanche americana* was found on only one of the 372 scaffolds taken into consideration based on the BLAST for pt features with low stringency settings. The exclusion of the other scaffolds was due to a combination of coverage and quality of the BLAST hit. Careful analysis showed that many of the hits, often only a few bp, were for features that are similar between the different plant genomes (e.g., tRNA, rRNA), especially between the mitochondrial genome and the plastome. An additional BLAST with mitochondrial features (data not shown) revealed overlapping with the pt BLAST results. As a result, the plastome of *Prosopanche americana* solely consists of scaffold 424 and shows a high similarity to the plastome of the sister genus *Hydnora* with respect to genome size, gene content and order as well as levels of AT richness. Both Hydnoraceae plastomes contain the identical gene set with the exception of two additional tRNA genes in *Prosopanche* (*trn*W, *trn*Y). All ribosomal RNAs and protein coding genes are putatively functional (based on feature continuous reading frames), with *rps*7 being the exception for *Hydnora visseri* and *rps*19 the exception for *Prosopanche americana*. The plastome of *Aristolochia*, the closest available, photoautotrophic relative, is about six times the size of Hydnoraceae and contains over six times as many genes with the complete set of plastid-encoded photosynthetic genes. These findings make Hydnoraceae ideal candidates to visualize and compare the degree of reduction and gene loss in a parasitic lineage that is close to a potential endpoint, the potential complete plastome loss [14]. Orobanchaceae, a relatively young parasitic lineage [29], display the entire range of parasitic lifestyles [19] and allow for the visualization of most of the proposed stages of plastome breakdown [5,16,17]. However, plastomes of Orobanchaceae are not nearly as reduced in size or gene content as some mycoheterotrophes and a few parasitic plants, such as *Pylostyles* [12,21], *Balanophora* [22] or Hydnoraceae. The quadripartite plastome structure with two single copy regions and inverted repeats is common between most photoautotrophic plants and also retained in the majority of the parasitic lineages. *Hydnora* [10] still possesses an IR, though only containing one complete gene (*trn*I-CAU) and two pseudogenes. With a length of 1466 bp, it is much smaller than the one of *Aristolochia contorta* (25,459 bp) [34] containing 17 complete genes and one pseudogene. *Prosopanche americana* also contains a gene duplication (*trn*I-CAU), but as both copies appear in the same orientation, this is not to be considered an inverted repeat. The fact that the duplicated gene as well as the partial gene (*rpl*2) on the repeats can be found in the inverted repeat of *Aristolochia* as well as the extremely reduced IR of *Hydnora visseri* leads to the conclusion that these duplicated regions are remnants of an inverted repeat and potentially have changed orientation during a rearrangement process that occured along with genome reduction in *Prosopanche americana.* The coverage in both duplicated regions in *Prosopanche* is also not significantly higher than in the rest of the plastome, something usually common for inverted repeats. Additionally, the sequencing reads representing both copies are not assembled together as one sequence, which is the case for inverted repeats due to their identical sequence. The loss of one IR copy in *Prosopanche* could also be associated with the larger of the two inversions (‘*trn*I-*rpl*2-*rps*2′) when comparing the gene order of the two Hydnoraceae genera. As these genes usually can be found in the inverted repeats, the loss of a different copy in *Prosopanche* compared to *Hydnora* could result in this apparent inverted gene order, as well as possible recombination events, for example, flip-flop recombination resulting in two plastome isoforms [44] before the IR loss in *Prosopanche*. The position of the second copy of the 195 bp duplication in the latter, nested in between the ribosomal RNA genes is noteworthy, as those are usually part of the inverted repeat. Therefore, the repeat position and the inverted region in comparison to *H. visseri* could potentially be a consequence of the IR loss, which is shown to lead to an increase in genome rearrangements. The occurrence of direct repeats instead of inverted repeats, although larger and containing more genes, is also known for species of the photoautotrophic genus *Selaginella* [45,46] and it is hypothesized that their recombination and gene conversion are not inhibited by the orientation of the direct repeat [47].

All protein coding genes present in the plastome of *Prosopanche americana* have a feature continuous reading frame, making them very likely functional (potential exception *rps*19) although evidence from transcriptome is currently unavailable. Functionality through structural analysis of the six tRNAs, as tested with tRNAscan-SE [38,39] is likely given for five of them. The sequence for *trn*W-CCA (Trp) is predicted by tRNAscan to have an ACA anticodon instead of a CCA anticodon. No tRNA with an ACA anticodon is known from Piperales plastomes but was found, for example, in Asterales (e.g., *Pogostemon cablin* MF287373 [48]). Sequence similarity strongly suggests that said Asterales tRNA with ACA anticodon is known as the intron-containing *trn*V-UAC in Piperales. Alignments of neither Asterales tRNA nor the *trn*V from Piperales did result in reasonable alignments.

Evaluation of nucleotide compositions of the shared protein coding genes between Hydnoraceae and *Aristolochia* visualizes the proportional increase in A and T nucleotides, also known from other plastomes of parasitic lineages [5,22]. *Hydnora* and *Prosopanche* showed a drastic decrease in G and C nucleotides across the whole plastome compared to the photoautotrophic *Aristolochia* with *Prosopanche* having slightly lower values than the sister genus *Hydnora*. Despite the drastically low proportion of G and C nucleotides in the Hydnoraceae plastomes compared to most photoautotrophic species, the record setting AT richness is known for holoparasitic *Balanophora* with most protein coding genes consisting to more than 90% of A and T [22]. For protein coding genes, mutations to the favored nucleotides predominantly occur on the third, variable codon position which has also been observed for other parasitic lineages such as Orobanchaceae [3]. Out of the three available stop codons (i.e., TAG, TGA and TAA) only two (TAG and TAA) are found in the protein coding genes of *Prosopanche*. The predominantly used stop codon found was TAA, highlighting another specific case of A and T nucleotide preference, whereas TAG was only found twice (*acc*D, *rpl*16) as stop codon. Similar cases of favoring certain stop codons or exclusively using a single one have been described, e.g., for the parasitic lineage *Cytinus* (Cytinaceae) [11] and *Balanophora* (Balanophoraceae) [22].

A comparison of the proportional amount that protein coding regions, ribosomal RNAs and IGRs contribute to the whole plastome highlights the many macro level changes Hydnoraceae plastomes experienced in comparison to the photoautotropic *Aristolochia*. The parasitic genomes have lost all plastid encoded genes related to the photosynthesis apparatus and their intergenic regions show a much more drastic decrease in length compared to *Aristolochia*, resulting in a proportional increase of space that the remaining protein coding genes and ribosomal RNAs take up in the plastome. 

The comparisons of the two Hydnoraceae plastomes and the highlighted common set and order of genes between the genomes that have evolved independently since the split ~54 MYA lead to two possible scenarios. Either a similarly reduced plastome existed already in the MRCA of *Hydnora* and *Prosopanche* and did not experience many changes with respect to gene content and order since or the observed similarities are a result of two independent evolutionary pathways leading to the same endpoint. Although the first hypothesis would be more parsimonious, a larger sampling within Hydnoraceae would be needed to test these hypotheses. However, the second hypothesis would support the existence of taxon dependent essential gene sets for heterotrophic plants [3,49].

## 4. Materials and Methods 

### 4.1. Plant Material, DNA Extraction, Library Preparation and Sequencing 

Plant material of *Prosopanche americana* was collected near Chancaní (Córdoba, Argentina). A herbarium voucher with number AAC5681 was placed at Museo Botánico de Córdoba (CORD). Fresh tepal material was sliced and air dried for 24 hours and subsequently stored in silica gel. DNA extraction was done using the DNeasy Plant Maxi Kit (Qiagen, Venlo, Netherlands). Molecular weight of isolated DNA as well as quality was tested using gel electrophoresis, and NanoDrop 2000 (Thermo Scientific, Waltham, MA, U.S.). Illumina TruSeq DNA PCR-free library preparation was done using 2.2 µg genomic DNA with an insert size of 650 bp. The library was sequenced using a partial lane of an Illumina NextSeq High Output with 300 cycles creating 37,324 million reads. Both, the library preparation and library sequencing were performed at the DRESDEN concept Genome Center of TU Dresden (https://genomecenter.tu-dresden.de). 

### 4.2. Assembly and Identification of Scaffolds with Plastid Information

Using the de novo assembly function, raw data were assembled with CLC Workbench [35] allowing for automatic word and bubble size. The assembly resulted in a total of 8,342,403 scaffolds to which the reads were mapped back to obtain coverage information and to be able to evaluate the assembly in specific regions of interest. The assembly was blasted (BLASTn, e-value e1-10) against a database containing 45 angiosperm plastid reference genomes from GenBank. Contigs with hits to plastid genomes were extracted and used together with the readmapping information for the creation of a stoichiometry plot in RStudio [50] allowing for highlighting of contigs with hits to plastid gene features followed by a directed and more efficient search for contigs potentially belonging to the plastid genome. The latter contigs were selected for further investigation.

### 4.3. Gene Annotation and Circularization

To check for and identify genes the scaffolds were uploaded and analyzed using DOGMA [36] at low stringencies (percent identity cutoff for protein coding genes and RNAs = 25, E-value = 1e-5) and additionally using MFannot [37]. Additionally, alignment tools such as “Map to reference” and LASTZ version 7.0.2 [51] were used as a plug in within Geneious [52] and applied to the scaffolds identified in the initial BLASTn search and the stoichiometry plot. In addition, the published plastome of *Hydnora visseri* (NC_029358) [10] was used to identify and annotate genes due to potentially higher sequence similarity between the sister genera. Gene boundaries were identified, then genes were annotated using Geneious [52] by aligning respective genes of *Hydnora visseri* and closely related photoautotrophic species of Piperales and in case of protein coding, genes feature continuous reading frames were checked manually. As a proxy of functionality of tRNA genes, in the absence of transcriptome data, the respective sequences were used as input for tRNAscan-SE [38,39] to predict their 2D structure. Criteria used were a cloverleaf structure with three leaves and that the predicted anticodon through tRNAscan found at the anticodon arm matches the anticodon predicted through sequence similarity. The full output is available as Appendix A.

The *Prosopanche americana* plastome was finally circularized using the “Circular Sequence” tool in Geneious [52] and drawn using OGDRAW [53]. The correctness of this circular connection was verified using PCR followed by Sanger sequencing (forward and reverse) with a forward (ProAm-rps2F: AACTAAATTACAAGCCATTGATA) and a reverse primer (ProAm-rps14R: TCCTAGAGGTTATTATCGTTAT) derived from the available flanking regions.

### 4.4. Plastome Comparisons

The plastome of *Prosopanche americana* (GenBank accession number MT075717) was compared with the plastome of its sister genus *Hydnora* (*Hydnora visseri*, NC_029358) [10] as well as a plastome of its close autotrophic relative *Aristolochia* (*Aristolochia contorta*, NC_036152) [34] in terms of multiple parameters such as length, gene content and order, GC content and plastome structure with Geneious [40]. A dotplot (Score matrix: exact, window size: 100, threshold: 200), implemented in Geneious, was used to visualize differences between the *Prosopanche* and *Hydnora* plastome nucleotide sequences. For the protein coding genes shared between the three species, the rates of the four nucleotides were determined using CodonW (version 1.4.2) [41].

### 4.5. Phylogenetic Placement of Prosopanche Americana

Protein coding and rRNA genes of *Prosopanche americana* and *Hydnora visseri* were added and aligned to the 81 angiosperm-wide alignments published by Jansen et al. [42]. Additionally, an alignment for the *acc*D gene was created using data from GenBank. The sampling was then improved by adding all complete plastid genomes of Piperales accessions available in GenBank (https://www.ncbi.nlm.nih.gov/), increasing the data set to 83 taxa. After automatic alignment with MAFFT (version 7.450) [54,55], individual alignments were inspected and improved by eye using AliView (version 1.20) [56] and concatenated in Geneious [40]. Single gene alignments have been uploaded to TreeBASE (http://purl.org/phylo/treebase/phylows/study/TB2:S25836). RAxML analysis [57] using the GTR+I+G model with 1000 bootstrap replications was carried out on the concatenated data set using the CIPRES Science Gateway [58] after calculating the optimal substitution model using jModelTest (version 2.1.7) [59]. Phylogenetic trees were rooted using the three Gymnosperms (*Cycas*, *Ginkgo*, *Pinus*) as outgroup. The output was visualized in TreeGraph 2 [60]. In addition to the 82 gene analysis, we used a gene set reduced to the ones that are present in the Hydnoraceae (Appendix A) and also did RAxML analysis based on amino acid alignments (translated with Geneious [40]), both on the complete (Appendix A) and reduced gene set (Appendix A).

## Figures and Tables

**Figure 1 plants-09-00306-f001:**
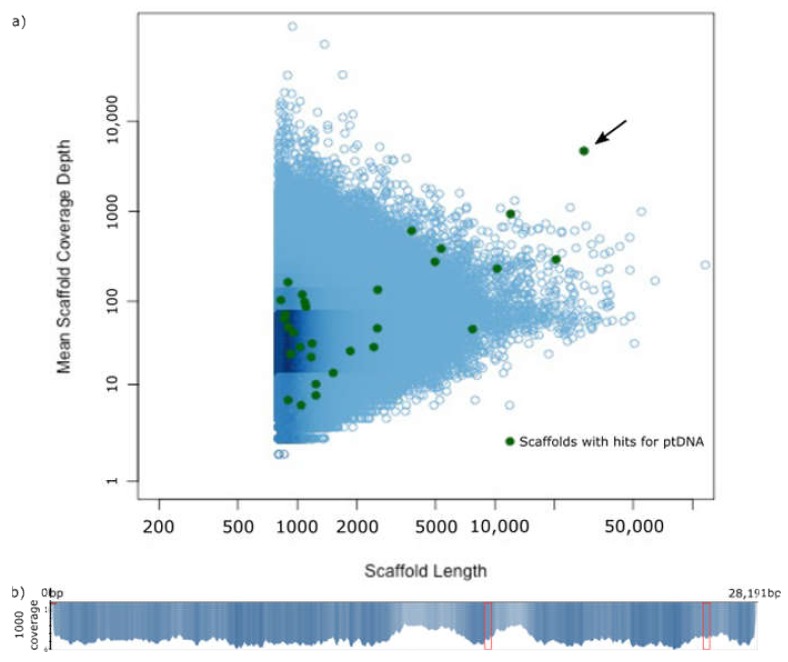
The plastid genome of *Prosopanche americana* is found on a single scaffold. **a**) Stoichiometry plot of the scaffold lengths and the respective mean coverage depth of the *Prosopanche americana* assembly. Scaffolds are visualized as blue circles. The green dots represent scaffolds with BLAST hits for annotated fractions of the plastome (derived from “gene features” in NCBI). The black arrow points to scaffold 424. **b**) Distribution of the reads mapped to the linear scaffold 424 with the scale displaying the depth of coverage in 1k increments. The red boxes represent the coverage of the *Prosopanche amercana* repeat regions.

**Figure 2 plants-09-00306-f002:**
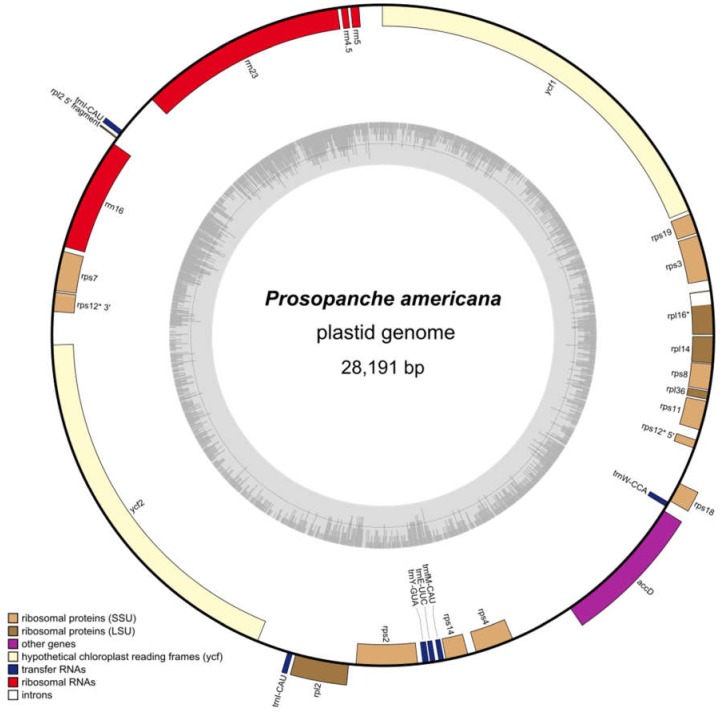
The highly reduced plastome of *Prosopanche americana*. The circular plastome of holoparasitic *Prosopanche americana* contains 24 unique genes (14 ribosomal protein genes, 4 ribosomal RNA genes, five genes coding for transfer RNAs and three genes with other or unknown function), which are distributed over a total length of 28,191 bp. Gene distribution is displayed on the outer circle with color coded gene groups according to the legend (bottom left). GC content is visualized as inner, grey circle. * indicates genes with introns.

**Figure 3 plants-09-00306-f003:**
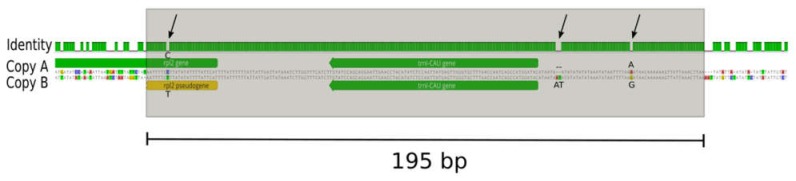
*Prosopanche americana trn*I-CAU repeat region. Alignment of the two *trn*I-CAU repeat copies highlights high sequence identity (green identity bar) over 195 bp with only four nucleotide differences (arrows pointing to gaps in identity graph and nucleotide differences highlighted by color and nucleotide code in sequence alignment). The *trn*I-CAU gene is fully contained within the repeat whereas only 25 bp of the *rpl*2 gene start fall within repeat borders. Contrary to copy B, which stays with a truncated *rpl*2 pseudogene, copy A is preceded by the complete *rpl*2 ORF of *Prosopanche americana*.

**Figure 4 plants-09-00306-f004:**
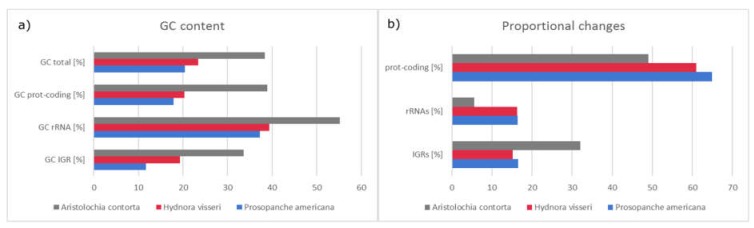
*Prosopanche americana* has a low GC content. (**a**) Comparison of the GC content and (**b**) proportional changes of plastid genome compartments (protein coding, ribosomal RNA, intergenic region) among the photosynthetic *Aristolochia contorta* (Aristolochiaceae, NC_036152) and the holoparasitic Hydnoraceae highlighting the extent of changes the plastomes of the Hydnoraceae have undergone with respect to nucleotide composition and conservation of specific genome regions.

**Figure 5 plants-09-00306-f005:**
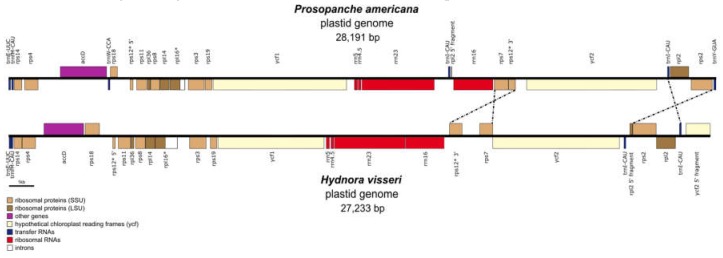
High structural similarity between the plastomes of Hydnoraceae. Linear plastid genome comparison of *Prosopanche americana* and *Hydnora visseri* (NC_029358). Genes are color coded; the dotted lines highlight inversions between the two genomes compared, genes containing introns are marked with *.

**Figure 6 plants-09-00306-f006:**
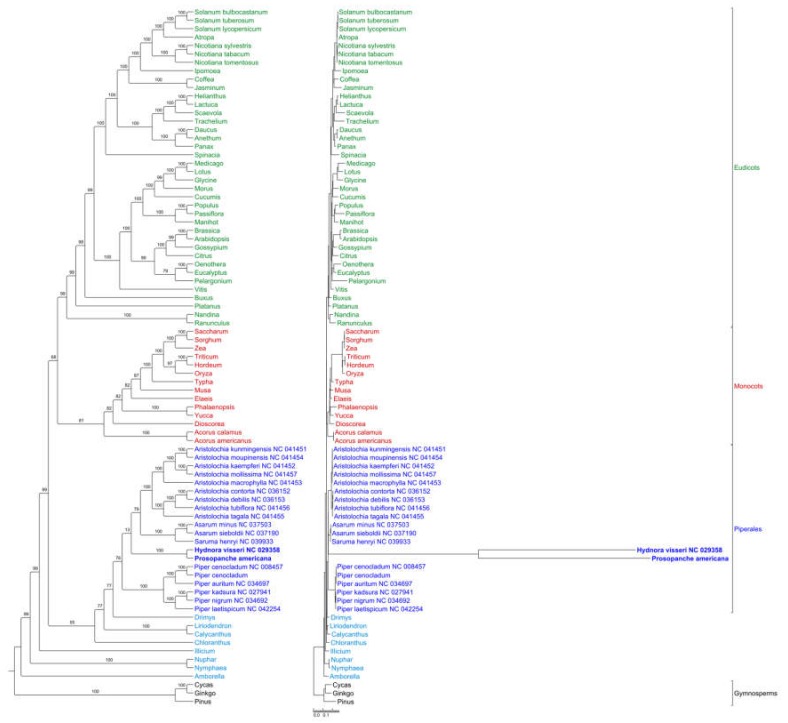
High mutational rate within Hydnoraceae relative to other Piperales and diverse photosynthetic angiosperms. Phylogenetic maximum likelihood tree reconstruction as cladogram (left) and phylogram (right) estimated using RAxML with the GTR + I + G model and conducting rapid bootstrapping (1000 replicates) recovers *Hydnora* and *Prosopanche* as sister genera and Hydnoraceae together as sister to a sister group of Aristolochioideae and Asaroideae (both Aristolochiaceae) within the Piperales. Bootstrap support values are displayed above the nodes. The scale bar shows the number of substitutions per site. Ingroup taxa are color coded, with eudicots being green, monocots red, Piperales dark blue and remaining Magnoliids and ANITA grade taxa light blue. Taxa refer to the dataset of Jansen et al. [42] if not otherwise indicated by GenBank number.

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
