# Peer review of "The First Plastid Genome of the Holoparasitic Genus Prosopanche (Hydnoraceae)"

_plants, 2020, doi:10.3390/plants9030306_

Round 1

Reviewer 1 Report

The manuscript entitled ‘The first plastid genome of the holoparasitic genus Prosopanche (Hydnoraceae)’ characterized the highly reduced plastid genome of Prosopanche americana. Additionally, the authors provide comparative analysis to the reduced plastid of Hydnora visseri, and a member of the Aristolochiaceae, closest photosynthetic relatives of these parasitic plants. Overall this is a well written manuscript that I believe could benefit from a few additional analyses, most importantly my third comment below, and small modifications to the text which are described below.

Comments:

372 scaffolds with putative ptDNA are identified and the only one further discussed is that which is described as the plastome. Were the other scaffolds investigated at all? Is it possible that multiple plastid chromosomes exist or are those evidence of plastid genes encoded on the nuclear genome? Mention these in the discussion at least and if they are not ptDNA, perhaps a better label can be used for the scaffolds marked in green in Figure 1.

Consider running the plastid genome annotation on MFannot (http://megasun.bch.umontreal.ca/cgi-bin/dev_mfa/mfannotInterface.pl) as well since DOGMA is being phased out by the software developers. "5/15/2019 DOGMA is no longer accepting new userids. The software is being sunsetted after 15 years and will not be availabe for use in the near future. I recommend using other newer plastid annotation tools. Sorry for the inconvenience." -https://dogma.ccbb.utexas.edu/ MFannot, in combination with the tRNAscan-SE results may help clarify some of the potential pseudogenes (tRNAs and rps19) that are discussed throughout the manuscript.

Please address the average depth of coverage in the plastome? The apparent loss of an inverted repeat can be an artifact of the CLC assembly. If the two regions containing tRNA-I and rpl2 contain significantly higher coverage than the rest of the palstome, it may be indicative of an improper assembly which has been observed in CLC previously. If this is the case, it may be able to be fixed following the methods described in Evans et al 2019. Chloroplast and mitochondrial genomes of Balbiania investiens (Balbianiales, Nemaliophycidae). Phycologia. DOI: 10.1080/00318884.2019.1573349. 

Running phylogenetic analysis using proteins and also removal of fast evolving site should help address the possibility of long branch attraction between Prosopanche and Hydnora. Also please make all alignments used in phylogenetic analyses publicly available through a repository (datadryad or something similar).

It would be beneficial for understanding the tRNA pseudo gene results if the full outputs from tRNAscan-SE were made available as supplementary material

Text modifications:

Abstract:

Line 13. Add a comma after ‘phototrophic sister species’

Introduction:

Line 47. ‘Extend should be changed to ‘extent’

Line 53. Change ‘that carry out functions other than photosynthesis related’ to ‘that carry out non-photosynthesis related functions’

Line 76. I would modify the end of the sentence to describe more specifically why it would be desirable. For understanding the evolution of parasitic plants? Reductive evolution of plastids? Something like that.

Results:

Line 86. Merge the two sentences that end and start with “scaffold 424” into a single sentence

Figure 4 Legend. Change ‘extend’ to ‘extent’

Reviewer 2 Report

References should be checked carefully.

E.g., 1. Annual review of genetics should be Annual Review of Genetics

  18.  Proceeding of National Academy of Sciences should be written in italic.

Reviewer 3 Report

Jost and colleagues present the extremely reduced cpDNA of the holoparasite Prosopanche americana (Hydnoraceae, Piperales) of only 28 KBp. Like in the previously reported cases of parasitic or mycoheterotrophic plants, the chloroplast genome is strongly impoverished in genes directly involved in photosynthesis that are otherwise highly conserved in green plants. The cpDNA of P. americana is highly similar to the one of the sister taxon Hydnora visseri previously determined. Hence, this manuscript does not reveal particularly exciting novel biological findings but rather adds another item to the collection of parasitic plant plastomes, in this case from an early diverging lineage among the flowering plants, the Piperales. The paper is clearly written although somewhat laden with less interesting text giving comparative numbers that could alternatively be summarized in a table and with some redundancy in discussion and results. I suggest some streamlining.

Other comments:

The authors initially identified many scaffolds with cpDNA homologies (ca. 30, with up to 10 KBp in length and significant coverage of ca. 1000) that they even display in a figure on its own, but they do not further comment on those in the following. Do they represent cpDNA integrated into the nuclear or mitochondrial DNA as frequently observed in plants? Such information could add value to the manuscript.

The discussion of the phylogenetic analyses is somewhat unclear. What is the intention of the extensive eudicot and monocot sampling? Concerning the placement of Hydnoraceae among the Piperales in question here, an alignment of a gene set reduced to the content of the Hydnoraceae cpDNAs would make sense and taxon sampling could be reduced to Piperales plus only a moderate collection of close outgroups. Note that a bootstrap value of 13 is not “low” (line 257), but rather simply no support at all.

In their considerations, the authors may wish to add information on the recently explored cpDNAs of Selaginella species that show striking variability of their chloroplast genomes including conversions of IRs intro DRs and the loss of ndh genes also in the absence of parasitic lifestyles, e.g. Selaginella tamariscina (Xu et al., 2018), S. kraussiana (Mower et al., 2018), S. vardei and S. indica (Zhang et al., 2019).

Minor issues:

Line 44: extenT

Line 121: trnY-GUA is discussed as a pseudogene owing to an “additional loop”. This, however, seems to be the variable extra arm between the anticodon and pseudouridine arm that is typical for trnY and some other tRNAs (Fig. S1). The authors should rather refer to the significant sequence divergence of trnY in Prosopanche compared to the sequences in other plant cpDNAs.

Line 303: The term “open reading frame” is generally used for potential protein coding regions of unknown function. I suggest to use the expression only in that sense but not for characterized genes. For example, the sentence starting line 303 should better read “feature continuous reading frames” than “have a open reading frame”.

The sequence of trnfM-CAU shows an “N” in position 2 (Fig. S1). Is this an unresolved sequence ambiguity. If yes, how many others remain in the assembled Prosopanche cpDNA?

Round 2

Reviewer 1 Report

I feel that the authors have sufficiently addressed the concerns I initially had with this manuscript. However, I think it would be useful for readers if the phylogenies produced from amino acid alignments and reduced gene sets were included in the supplemental material rather than not shown at all.